# Synthesis and Characterization of Pyrazole-Enriched Cationic Nanoparticles as New Promising Antibacterial Agent by Mutual Cooperation

**DOI:** 10.3390/nano12071215

**Published:** 2022-04-05

**Authors:** Silvana Alfei, Guendalina Zuccari, Debora Caviglia, Chiara Brullo

**Affiliations:** 1Department of Pharmacy, University of Genoa, Viale Cembrano, 16148 Genoa, Italy; zuccari@difar.unige.it (G.Z.); brullo@difar.unige.it (C.B.); 2Department of Surgical Sciences and Integrated Diagnostics (DISC), University of Genoa, Viale Benedetto XV, 6, I-16132 Genova, Italy; debora.cavigliad@edu.unige.it

**Keywords:** polystyrene-based cationic copolymer (P7), physical entrapment, 1-(3,5-diphenyl-1H-pyrazol-1-yl)-3-(isopropyl amino) propan-2-ol) hydrochloride salt (CB1H), water-soluble CB1H-P7 NPs, spherical morphology, triphasic release profile, Weibull mathematical kinetic model

## Abstract

A pyrazole derivative (CB1) was previously evaluated in vivo for various pharmacological activities (with the exception of antimicrobial effects), using DMSO as the administrative medium, mainly due to its water insolubility. Considering the global necessity for new antimicrobial agents, CB1 attracted our attention as a candidate to meet this need, mainly because the secondary amine group in its structure would make it possible to obtain its hydrochloride salt (CB1H), thus effortlessly solving its water-solubility drawbacks. In preliminary microbiologic investigations on Gram-negative and Gram-positive bacteria, CB1H displayed weak antibacterial effects on MDR isolates of Gram-positive species, nonetheless better than those displayed by the commonly-used available antibiotics. Therefore, aiming at improving such activity and extending the antibacterial spectrum of CB1H to Gram-negative pathogens, in this first work CB1 was strategically formulated in nanoparticles using a cationic copolymer (P7) previously developed by us, possessing potent broad-spectrum bactericidal activity. Using the nanoprecipitation method, CB1H-loaded polymer nanoparticles (CB1H-P7 NPs) were obtained, which were analyzed by attenuated total reflection-Fourier transform infrared (ATR-FTIR) spectroscopy to confirm the successful loading. Additionally, CB1H-P7 NPs were fully characterized in terms of morphology, size, polydispersity indices, surface charge, DL%, and EE%, as well as release and potentiometric profiles.

## 1. Introduction

Among heterocyclic compounds, the pyrazole ring represents one of the most significant chemical skeletons in medicinal chemistry and advanced organic materials [1]. In fact, the pyrazole derivatives possess several pharmacological properties, including among others anticancer, analgesic, anti-inflammatory, antidepressant, and beneficial cardiovascular effects [2].

It is commonly recognized that the pharmacological activity of pyrazole is due to its chemical characteristics, specifically to the presence of two nitrogen atoms with very different chemical properties, and to the planar structure of the aromatic heterocycle [3,4,5]. Particularly, the pyrazole nucleus, which is also present in natural compounds, contains three carbon atoms and two adjacent nitrogen atoms [6]. While the “pyrrole-like” nitrogen atom 1 (N1) possesses unshared electrons conjugated with the aromatic system, the “pyridine-like” nitrogen atom 2 (N2) is characterized by unshared electrons not compromised by resonance, similar to pyridine systems. Due to the chemical differences between N1 and N2, pyrazoles can react both with acids and bases [7]. Currently, the pyrazole ring is present in numerous commercialized drugs belonging to different therapeutic categories, including potent anti-inflammatory compounds acting by COX-2 inhibition (celecoxib), nonsteroidal anti-inflammatory drugs (NSAIDs) (tepoxalin), anticancer agents (crizotimib), anti-obesity agents (surinabant, difenamizole), tranquilizers (mepiprazole), insecticides (finopril), antipsychotics (CDPPB), analgesics (betazole), and H2-receptor agonists and antidepressant agents (fezolamide), thus confirming the pharmacological potential of the pyrazole nucleus [4,5]. To meet the joint initiative of the World Health Organization (WHO) and Drugs for Neglected Diseases initiative (DNDI), which encourages research and development through public–private partnerships of new antibiotic drugs [8], we are particularly interested in pyrazole-containing molecules possessing antibacterial effects.

Very recently, dendrimer nanoparticles (NPs) loaded with a 3,5-diphenyl pyrazole derivative, namely BBB4 (BBB4-G4K NPs), designed to solve the solubility drawbacks of BBB4 [9], which is otherwise water-insoluble and not clinically applicable, demonstrated potent antibacterial effects, specifically against *Staphylococci* [10]. In particular, BBB4-G4K NPs were active on methicillin-resistant *Staphylococcus aureus* (MRSA) and *S. epidermidis* (MRSE), also resistant to linezolid. It should be noted that these pathogenic species are particularly difficult to treat, due to their resistance to antibiotics and responsibility for different types of infections, including skin infections, bacteremia, bone infections, endocarditis, food poisoning, pneumonia, and toxic shock syndrome (TSS), a life-threatening condition caused by toxins which are often resistant to antibiotics [10].

Although novel antibacterial agents effective against multidrug-resistant (MDR) bacteria of Gram-positive species are desired, the Antimicrobial Availability Task Force of the Infectious Diseases Society of America, the FDA, and other organizations have highlighted the urgent need to develop new antibiotics with activity against Gram-negative organisms [11].

Hence, with the aim of developing other new pyrazole-based antibacterial agents with a hopefully broader spectrum of action, and active also against MDR Gram-negative pathogens, we studied an in-house pyrazole library constructed over the years, including compounds synthesized by Bruno et al. in the 1990s [12]. Among the other pyrazoles, compound CB1 [1-(3,5-diphenyl-1H-pyrazol-1-yl)-3-(isopropyl amino) propan-2-ol)] (Figure 1), previously evaluated in vivo using DMSO as means of administration due to water insolubility, had showed some biological activities, such as analgesic, antiaggregating and antiarrhythmic effects [12].

Interestingly, no studies have ever been conducted to investigate the possible antibacterial effects of CB1, mainly because of its insolubility in water, which made it difficult to read the values of minimum inhibitory concentration (MICs) correctly and reliably. However, given the results obtained with the structurally similar BBB4, we felt that CB1 could be an excellent candidate as a new antibacterial agent. As well, the structure of CB1 attracted our interest because, in addition to possessing two phenyl groups on C3 and C5 and a hydroxyethyl chain on N1 as BBB4, it possessed a secondary amine group. The presence of this basic group allowed us to prepare the water-soluble hydrochloride salt of CB1 (CB1H), thus solving in advance its water-solubility drawbacks, and permitting in vitro microbiological evaluation in aqueous media, and future in vivo administration without using DMSO.

Then, since preliminary antimicrobial investigations with CB1H on representative MDR isolates of Gram-negative and Gram-positive species showed weak antibacterial activity exclusively on bacteria of Gram-positive species, in this work we aimed at developing a tactic to improve and broaden its activity. Generating strategies for the delivery of hydrophilic drugs is emerging as an important research field, and the entrapment of hydrophilic small molecules in macromolecular systems has offered many advantages, such as their protection against in vivo degradation, the reduction of potentially toxic side effects, and the enhancement of their efficacy [13].

Among the different nanocarriers, nanoparticles, including both nanocapsules and nanospheres, have emerged as a very promising strategy [14].

In this scenario, aiming at enhancing the antibacterial properties of CB1H and at extending its spectrum of action to Gram-negative bacteria, we decided to strategically nano-formulate it using a cationic copolymer (P7), recently reported by us, possessing potent broad-spectrum bactericidal activity [15]. Additionally, the cationic nature of P7 would promote the interaction of the CB1H-loaded P7 nanoparticles we aimed to prepare with the negative bacterial surface, thus favoring their localization and accumulation on prokaryotic cells. Moreover, it is now generally accepted that cationic macromolecules, thanks to electrostatic interactions with the bacterial surface, cause the depolarization of the membrane and its progressive permeabilization through the formation of increasingly large pores [16,17]. In our case, pore formation would have favored the entry of CB1H into the bacterial cell, thus promoting its interference with specific metabolic pathways of both Gram-positive and Gram-negative bacteria. So, in this first work based on CB1, CB1H was physically entrapped in P7 by a nanoprecipitation process without the aid of harmful additives or high-boiling-point solvents difficult to remove completely. CB1H-loaded polymer nanoparticles (CB1H-P7 NPs) were obtained, which were first analyzed by attenuated total reflection-Fourier transform infrared (ATR-FTIR) spectroscopy to confirm the successful loading. The obtained CB1H-P7 NPs were subsequently fully characterized in terms of morphology, size, polydispersity indices, surface charge, DL%, EE%, and release profiles. Potentiometric titrations were also performed to investigate the protonation profile of CB1H-P7 NPs at physiological pH, in order to confirm the subsistence of a cationic character, essential for exerting antibacterial activity in such conditions. Principal component analysis (PCA) was also exploited to process the ATR-FTIR spectral data of CB1H, P7, and CB1H-P7 NPs, obtaining reliable predictive information about the chemical composition of the prepared CB1H-P7 formulation.

## 2. Material and Methods

### 2.1. Chemicals and Instruments

The polystyrene-based random copolymer (P7) used for entrapping the pyrazole derivative CB1H was prepared according to a previously described procedure [18], starting from the monomer 2-methoxy-6-(4-vinylbenzyloxy)-benzylammonium hydrochloride (M7). M7 was in turn prepared following a multi-step procedure described in our previous work [18]. The main features, the ATR-FTIR and NMR data, as well as the elemental analysis results of M7 and P7, have been reported in Section 2.2. All reagents and solvents were purchased from Merck (formerly Sigma-Aldrich, Darmstadt, Germany), and were reagent grade. Solvents were purified by standard procedures, whereas reagents were employed as such, without further purification. Melting points are uncorrected. NMR spectra for ^1^H and ^13^C were acquired on a Bruker DPX spectrometer (Bruker Italia S.r.l., Milan, Italy), at 300 and 75.5 MHz respectively, or on a Jeol 400 MHz spectrometer (JEOL USA, Inc., Peabody, MA, USA) at 400 and 100 MHz, respectively. Fully decoupled ^13^C NMR spectra were reported. Chemical shifts were reported in ppm (parts per million) units relative to the internal standard tetramethylsilane (TMS = 0.00 ppm), and the splitting patterns were described as follows: s (singlet), d (doublet), t (triplet), q (quartet), m (multiplet), and br was added for broad signals. Centrifugations were performed on an ALC 4236-V1D centrifuge at 3400–3500 rpm. Elemental analyses were performed on an EA1110 Elemental Analyzer (Fison Instruments Ltd., Farnborough, Hampshire, England). Scanning electron microscopy (SEM) images were obtained with a Leo Stereoscan 440 instrument (LEO Electron Microscopy Inc., Thornwood, NY, USA). Dynamic Light Scattering (DLS) and Z-potential (ζ-p) determinations were performed using a Malvern Nano ZS90 light-scattering apparatus (MalvernInstruments Ltd., Worcestershire, UK). Potentiometric titrations were performed with a Hanna Micro-processor Bench pH Meter (Hanna Instruments Italia srl, Ronchi di Villafranca Padovana, Padova, Italy). Lyophilizations were performed using a freeze-dry system (Labconco, Kansas City, MI, USA). The obtained NPs were sonicated using a Bandelin SONOREX™ SUPER RK 52H instrument, with built-in heating ultrasonic baths with a capacity of 1.8 L, HF-Frequency 35 kHz (Bandelin Electronic GmbH & Co. KG, Berlin, German). A thin layer chromatography (TLC) system employed aluminum-backed silica gel plates (Merck DC-Alufolien Kieselgel 60 F254, Merck, Washington, DC, USA), and detection of spots was made by UV light (254 nm), using a handheld UV Lamp, LW/SW, 6W, UVGL-58 (Science Company^®®®^, Lakewood, CO, USA). Organic solutions were dried over anhydrous magnesium sulphate and were evaporated using a rotatory evaporator operating at a reduced pressure of about 10–20 mmHg.

### 2.2. ATR-FTIR, NMR and Elemental Analysis of M7 and P7

M7. Overall yield of 43%. Mp. 215–218 °C (acetonitrile). Purity 99% by HPLC. ATR-FTIR (ν, cm^−1^) 3500–3000 (NH_3_^+^); 1599, 1474 (N-H bending of NH_3_^+^) 999 and 923 (CH_2_=CH). ^1^H NMR (CD_3_OD, 300 MHz, δ ppm) 3.90 (s, 3H), 4.20 (s, 2H); 5.16 (s, 2H); 5.23 (dd, 1H, *J _gem_* = 1.0 Hz; *J _cis_* = 10.9 Hz); 5.82 (dd, 1H, *J _gem_* = 1.0 Hz; *J _trans_* = 17.6 Hz); 6.70–6.80 (m, 3H); 7.33–7.45 (m, 5H). ^13^C NMR (75.5 MHz, δ ppm) 33.41, 56.45, 71.50, 105.06, 106.50, 110.26, 114.46, 127.43, 128.95, 132.46, 137.70, 137.73, 138.95, 159.05, 160.28. Anal. Calcd. for C_17_H_20_ClNO_2_: C, 66.77%; H, 6.59%; N, 4.58%; Cl, 11.59%. Found: C, 66.80%; H, 6.58%; N, 4.60%; Cl, 11.62%.

P7. FTIR (ν, cm^−1^) 2500–3600 (NH_3_^+^); 2926 (C-H stretching alkyl groups); 2051, 1980 (aromatic overtones); 1618 (C=ONH); 1598, 1476 (N-H bending of NH_3_^+^); 12,591,129 (C-O).

### 2.3. Synthesis of 1-(3,5-diphenyl-1H-pyrazol-1-yl)-3-(isopropylamino)propan-2-ol Hydrochloride (CB1H)

First, 3,5-diphenyl-1H-pyrazol-1-yl-3-(isopropyl amino) propan-2-ol (CB1) was synthetized, slowly modifying the previously reported method [12] (Section 2.3.1 and Section 2.3.2); then CB1 was converted to its hydrochloride salt (Section 2.3.3).

#### 2.3.1. Synthesis of 1-(oxiran-2-ylmethyl)-3,5-diphenyl-1H-pyrazole (B)

3,5-diphenyl-1H-pyrazole (A) (Fluorochem Ltd. Hadfield, Derbyshire, United Kingdom), (220 mg, 1 mmol) and epichlorohydrin (0.16 mL, 1.5 mmol) were cooled at 5 °C. Afterward, triethylamine (TEA) (2.7 mmol, 0.2 mL) was added dropwise, and the reaction mixture was stirred until the temperature reached 25 °C. Then, the mixture was heated at 70 °C for 1 h. After cooling to room temperature, the mixture was poured into water (100 mL), and the aqueous phase was extracted with CH_2_Cl_2_ (2 × 10 mL). The organic phase was washed with water (10 mL) and brine (3 × 10 mL), dried (MgSO_4_), and concentrated under reduced pressure, yielding a yellow oil that was purified by silica gel (100–200 mesh) column chromatography, using diethyl ether as eluent. The oil was used without any additional purification.

#### 2.3.2. Synthesis of 1-(3,5-diphenyl-1H-pyrazol-1-yl)-3-(isopropyl amino) propan-2-ol

A solution of 1-(oxiran-2-ylmethyl)-3,5-diphenyl-1H-pyrazole (B) (276 mg, 1 mmol) in isopropyl amine (2 mL, mmol) was heated at reflux overnight. After cooling to room temperature, the mixture was poured into water (100 mL), and the aqueous phase was extracted with ethyl acetate (2 × 10 mL). The organic phase was washed with water (10 mL) and brine (3 x 10 mL), dried (MgSO_4_), and concentrated under reduced pressure, yielding an oil that was crystallized by ethyl ether (Et_2_O) as a white solid. Yield: 50 %. Mp. 115–116 °C.

^1^H-NMR (DMSO-*d6,* 400 MHz, δ ppm) 0.92 (s, 6H, 2 CH_3_); 2.44–2.72 (m, 3H, *CH*NH + *CH_2_*NH); 3.48 (1H, exchangeable with D_2_O); 4.04–4.26 (m, 3H, CH_2_N + *CH*OH); 6.83 (s, 1H, H-3); 7.20–7.88 (m, 10H, phenyl rings). ^13^C-NMR (DMSO-*d6,* 100 MHz, δ ppm) δ 150.26, 149.92, 146.04, 145.98, 141.34, 133.80, 133.57, 130.86, 130.54, 129.58, 129.19, 128.64, 125.62, 103.66, 69.16, 53.69, 51.32, 50.74, 22.73, 22.58. Anal. Calcd. for C_21_H_25_N_3_O: C, 75.19%; H, 7.51%; N, 12.53%. Found: C, 75.00%; H, 7.44%; N, 12.20%.

#### 2.3.3. CB1 Hydrochloride (CB1H)

CB1 was dissolved in anhydrous DCM (10 mL), and the corresponding hydrochloride salt was obtained by adding an ethyl ether HCl saturated solution (C_22_H_25_N_3_OCl). Mp. 52–55 °C.

ATR-FTIR (ν, cm^−1^): 3174 (OH), 3057 (stretching CH aromatics); 2984, 2924, 2843 (stretching CH alkyls); 2756, 2707, 2425 (NH_2_^+^ stretching); 1614, 1586, 1475 (stretching C=C, C=N and N-H bending of NH_2_^+^); 1108 (C-O.)

^1^H-NMR (DMSO-*d6,* 400 MHz, δ ppm) 1.17 (s, 6H, 2CH_3_); 2.86–3.29 (m, 3H, *CH*NH + *CH_2_*NH); 4.14–4.44 (m, 4H, CH_2_N + *CH*OH + 1H, exchangeable with D_2_O); 6.85 (s, 1H, H-3); 7.26–7.61 (m, 10H, phenyl rings); 8.60 (br s, 1H, exchangeable with D_2_O); 8.91 (br s, 1H, exchangeable with D_2_O).

### 2.4. Microbiological Screening

A total of 18 strains of Gram-negative and Gram-positive species were utilized in this work to screen the antibacterial properties of CB1H. All bacteria were clinical isolates, part of a collection provided by the School of Medicine and Pharmacy of the University of Genoa (Italy). They were identified by VITEK^®®®^ 2 (Biomerieux, Firenze, Italy) or matrix-assisted laser desorption/ionization time-of-flight (MALDI-TOF) mass spectrometric technique (Biomerieux, Firenze, Italy).

Three strains were of Gram-negative species (1 *Escherichia coli*, 1 MDR *Klebsiella pneumoniae*, and 1 MDR *Pseudomonas aeruginosa*), while 15 strains were of Gram-positive species. Among Gram-positive isolates, 8 were of *Enterococcus* genus, including 4 *E. faecalis* (3 vancomycin-resistant (VRE) and 1 vancomycin-susceptible (VSE)) and 4 *E. faecium* (2 VRE and 2 VSE); 7 were methicillin-resistant clinical isolates of the genus *Staphylococcus*, including 4 methicillin-resistant (MRSA) *S. aureus* and 3 methicillin-resistant (MRSE) *S. epidermidis*, including 2 isolates also resistant to linezolid.

To screen the antibacterial effects of CB1H, we determined its MICs, following the microdilution method according to the European Committee on Antimicrobial Susceptibility Testing (EUCAST) [19] and as previously reported [10].

### 2.5. Experimental Procedure to Prepare CB1H-Loaded P-Based NPs (CB1H-P7 NPs)

The CB1H-P7 NPs were prepared using the nanoprecipitation technique [20]. Briefly, a methanol (MeOH) 1/1 wt/wt solution of P7 and CB1H was prepared, dissolving P7 (84.4 mg, 0.00615 mmol) and CB1H (83.8 mg, 0.2253 mmol) in 1 mL MeOH. Et_2_O (10 mL) was used as non-solvent. The clear solution of the two ingredients (P7 and CB1H) was added to the non-solvent phase dropwise, using a Pasteur pipet, at room temperature and under moderate magnetic stirring (500 rpm), obtaining a fine white precipitate, which was kept under vigorous stirring overnight. The suspension was sonicated for 10 min at 25 °C twice, and the precipitate was recovered by centrifugation at 3500 rpm for 15 min. Once separated from the supernatant, the solid was resuspended in Et_2_O, and the suspension was sonicated for 10 min at 25 °C and centrifugated again at 3500 rpm for 15 min. The solid was separated from the solvent, and it was brought to constant weight at reduced pressure, obtaining 140.5 mg of an amorphous white solid, which was stored in a dryer on P_2_O_5_. The organic solvent was evaporated using a Rotavapor^®®®^ R-3000 (Büchi Labortechnik, Flawil, St. Gallen, Switzerland) to recover unentrapped CB1H, and its structure was confirmed by ATR-FTIR analyses (data not reported).

ATR-FTIR of CB1H-P7 NPs (ν, cm^−1^): 3174 (OH of CB1H), 2983, 2926, 2839 (stretching CH alkyls of CB1H and P7), 2756, 2706, 2423 (NH_2_^+^ stretching of CB1H and NH_3_^+^ stretching of P7), 1613, 1586, 1475 [stretching C=C, C=N (CB1H), N-H bending of NH_2_^+^ (CB1H) and NH bending of NH_3_^+^(P7)], 1108 (C-O of CB1H).

### 2.6. Chemometric Assisted ATR-FTIR Spectroscopy

FTIR spectra of CB1H-P7 NPs, CB1H, and P7 were recorded directly on the solid samples in attenuated total reflection (ATR) mode using a Spectrum Two FT-IR Spectrometer (PerkinElmer, Inc., Waltham, MA, USA). Acquisitions were made in triplicate from 4000 to 600 cm^−1^, with 1 cm^−1^ spectral resolution, co-adding 32 interferograms, with a measurement accuracy in the frequency data at each measured point of 0.01 cm^−1^, due to the internal laser reference of the instrument. The “find peaks” command of the instrument software was used to obtain the frequency of each band. The matrix of spectral data was subjected to PCA using PAST statistical software, (paleontological statistics software package for education and data analysis, free and downloadable online at: https://past.en.lo4d.com/windows; accessed on 19 February 2022). In particular, we arranged the FTIR data of the spectra acquired for CB1H, P7 and CB1H-P7 NPs in a matrix 3401 × 3 (*n* = 10,203) of measurable variables. For each sample, the variables consisted of the values of absorbance (%) associated to the wavenumbers (3401) in the range 4000–600 cm^−1^.

### 2.7. Content of CB1H in CB1H-P7 NPs, Drug Loading (DL%) and Entrapment Efficiency (EE%)

First, the CB1H calibration curve was constructed. A stock solution was prepared, dissolving CB1H (0.5926 mg/mL) in MeOH and, upon dilutions with MeOH, standard solutions at concentrations of 0.002963, 0.005926, 0.011852, and 0.023704 mg/mL were prepared. The CB1H contained in each solution was determined using a UV-Vis instrument (HP 8453, Hewlett Packard, Palo Alto, CA, USA) supplied with a 3 mL cuvette, by detecting the absorbance ([A]) at room temperature and at ʎ_abs_ = 250 nm. The CB1H concentrations were reported in a dispersion graph vs. the [A] values, and the CB1H calibration curve was obtained by least-squares linear regression analysis. Determinations were made in quadruplicate, and the [A] values obtained for each CB1H concentration analyzed were expressed as mean ± SD. The equation of the developed linear calibration model was the following Equation (1),
y = 67.121x + 0.0503(1)
where y was the [A] values measured at ʎ_abs_ = 250 nm and x the CR232 standard concentrations analyzed. In Equation (1), the slope represents the coefficient of extinction (ε) of CB1H.

To estimate the CB1H content in CB1H-P7 NPs, 3.442 mg of CB1H-P7 NPs were dissolved in 50 mL of MeOH (68.8 µg/mL), obtaining a stock solution which was vigorously stirred for ten minutes to promote the release of CB1H. The initial solution was further diluted to obtain four solutions at different concentrations (6.9, 13.8, 17.2, and 34.4 µg/mL). The amount of CB1H in the sample was quantified at ʎ_abs_ = 250 nm by UV-Vis analysis, using the same apparatus and the same conditions described in the previous section. The solutions were analyzed against a solution of the empty copolymer P7, as blank.

The values of DL% and EE% of CB1H-P7 NPs were calculated from the following Equations (2) and (3).
(2)$$DL\;\left(\%\right)=\frac{\mathrm{weight}\;\mathrm{of}\;\mathrm{the}\;\mathrm{drug}\;\mathrm{in}\;\mathrm{NPs}}{\mathrm{weight}\;\mathrm{of}\;\mathrm{the}\;\mathrm{NPs}}\times 100$$
(3)$$EE\;\left(\%\right)=\frac{\mathrm{weight}\;\mathrm{of}\;\mathrm{the}\;\mathrm{drug}\;\mathrm{in}\;\mathrm{NPs}}{\mathrm{inizial}\;\mathrm{amount}\;\mathrm{of}\;\mathrm{drug}}\times 100$$

The values of DL% obtained by UV-Vis analyses were used to estimate the average molecular mass (Mn) of CB1H-P7 NPs, consistent with previously reported procedures [9,21,22].

### 2.8. Morphology and Dynamic Light Scattering (DLS) Analysis of Particles of P7 and CB1H-P7 NPs

The morphology and average size of P7 and CB1H-P7 NPs were investigated by scanning electron microscopy (SEM), according to a previously described procedure [9,21,22]. The micrographs were recorded digitally using the DISS 5 digital image acquisition system (Point Electronic GmbH, Halle, Germany).

To determine the particle size (in nm), polydispersity index (PDI), and zeta potential (ζ-p) (mV) of P7 and CB1H-P7 NPs, we performed DLS analyses using a Malvern Nano ZS90 light-scattering apparatus (Malvern Instruments Ltd., Worcestershire, UK) and following our previously described procedure [9,21,22].

### 2.9. Potentiometric Titration of P7 and CB1H-P7 NPs

Potentiometric titrations were performed at room temperature, and the titration curves of P7 and CB1H-P7 NPs were obtained. The samples (20–30 mg) were dissolved in 30 mL of Milli-Q water (m-Q), then were treated with a standard 0.1 N NaOH aqueous solution [1.5 mL, pH = 9.64 (P7) and 9.84 (CB1H-P7 NPs)]. The solutions were potentiometrically titrated by adding 0.2 mL aliquots of a standard 0.1 N HCl aqueous solution, up to a total 3.0 mL, and measuring the corresponding pH values [23]. Titrations were made in triplicate and the determinations were reported as mean ± SD.

### 2.10. In Vitro CB1H Release Profile from CB1H-P7 NPs

The release profile of CB1H from CB1H-P7 NPs was investigated in vitro using the dialysis bag diffusion method. An exactly weighted quantity of CB1H-P7 NPs (10 mg) was dissolved in 1 mL of 0.1 M phosphate-buffered saline (PBS, pH = 7.4). The solution was then placed into a pre-swelled T2 tubular cellulose dialysis bag (flat width = 10 mm, wall thickness = 28 µm, V/cm = 0.32 mL) with a nominal molecular weight cut-off (MWCO) of 6000–8000 Da (Membrane Filtration Products, Inc., Seguin, TX, USA) and dipped into 20 mL of 0.1 M PBS, pH 7.4, at 37 °C with gentle stirring for 24 h. At predetermined time intervals (1 h, 2 h, 3 h, 4 h, 5 h, 6 h, 8 h, 10 h, 12 h, 24 h), 1 mL was withdrawn from the incubation medium and was analyzed by UV-Vis using the previously described apparatus. The exact amount of CB1H present in the samples was quantified at 250 nm, and the results were reported as the mean ± SD of three determinations. After sampling, an equal volume of fresh PBS was immediately replaced into the incubation medium.

The concentration of CB1H released from CB1H-P7 NPs was expressed as a cumulative release percentage (%) of the total amount of CB1H present in the CB1H-P7 NPs (according to the DL% value). The cumulative releases (%) of CB1H were plotted as a function of time, obtaining the CB1H release profile curve.

### 2.11. Statistical Analysis

The statistical significance of the slope of the CB1H calibration curve was investigated through the analysis of variance (ANOVA), performing the Fischer test. Statistical significance was established at the *p*-value < 0.05.

## 3. Results and Discussion

### 3.1. Monomer M7 and Copolymer P7

The monomer 2-methoxy-6-(4-vinylbenzyloxy)-benzylammonium hydrochloride (M7) was prepared following the multi-step procedure reported in Figure 1 [18].

The procedure shown in Figure 1 has been recently discussed [18].

Once M7 was obtained, copolymer P7 was obtained by the simple and low-cost radical copolymerization of M7 in solution with N,N-dimethylacrylamide (DMAA) in MeOH, using 2,2′-azobisisobutyronitrile (AIBN) as radical initiator, at 60 °C, achieving a conversion of 85% as described in Figure 2 [18]. Since preliminary investigations to assess the behavior of M7 in polymerization reactions have shown that M7 homopolymerized and copolymerized well in the abovementioned conditions, we avoided more complex polymerization procedures, such as atom transfer radical polymerization (ATRP) and reversible addition-fragmentation chain transfer (RAFT),

The experimental data of the copolymerization of M7, as well as the Mn and the NH_2_ equivalents contained in P7, are available in a recently reported work [18].

P7 was purified by repeated cycles of dissolution/precipitation using MeOH as solvent and Et_2_O as non-solvent. In the ATR-FTIR spectrum of P7 (Figure 2), bands deriving both from M7 and DMAA were observable. For M7, broad bands in the range 2500–3600 cm^−1^ due to the N-H stretching of the NH_3_^+^ groups, two intense bands at 1598 and 1476 cm^−1^ (N-H bending of NH_3_^+^), and strong bands at 1259, 1129 cm^−1^ due to the stretching C-O, were observable, while the contribution of DMAA was assessed by the presence of a strong band at 1618 cm^−1^ (C=ONH).

P7 was soluble in water, methanol, DMSO, and DMF, while insoluble in petroleum ether, diethyl ether and acetone.

### 3.2. Synthesis of CB1H

The compound CB1, 3,5-diphenyl-1H-pyrazol-1-yl-3-(isopropyl amino) propan-2-ol, was prepared according to Figure 3 [12].

Briefly, the commercially available pyrazole derivative A was reacted with epichlorohydrin to achieve the oxirane derivative B, which, without purification, was promptly converted to CB1 by reaction with isopropyl amine. The structure of CB1 was confirmed by ATR-FTIR, ^1^H and ^13^C NMR spectra, and elemental analysis. Subsequently, to obtain CB1 in a water-soluble form, its hydrochloride salt (CB1H) was prepared using a HCl saturated ethyl ether solution. As reported in the experimental section, the structure of CB1H was confirmed by ATR-FTIR and ^1^H NMR analyses. The ATR-FTIR and the ^1^H NMR spectrum (DMSO-*d6*) of CB1H are shown in the following Figure 3 and Figure 4, respectively.

Notably, the ATR-FTIR spectrum of CB1H presents the typical bands of the pyrazole nucleus. As expected, the spectrum was like that of BBB4 [9], with additional bands at 2756, 2707, and 2425 cm^−1^ due to the N-H bending of the NH2+ group. Moreover, the ^1^H NMR spectrum of CB1H showed that, although the pyrazole ring also contains a protonable basic nitrogen atom, in the formation of the hydrochloride salt, only the basic nitrogen atom of the secondary aliphatic amine group of CB1 underwent protonation, as confirmed by the two broad signals at 8.60 and 8.91 ppm integrable for two magnetically but not chemically equivalent proton atoms. Moreover, the very broad signal at 4.27 ppm, surely the signal of the OH group, denotes a significant presence of water, thus indicating the highly hygroscopic status of CB1H.

### 3.3. Screening of the Antibacterial Properties of CB1H

Hopeful that CB1H could be a good candidate as a new antibacterial agent, we examined its antibacterial activity on both Gram-positive and Gram-negative isolates, including MDR strains. In particular, we determined the MICs of CB1H on seven isolates of the *Staphylococcus* genus, on eight isolates of the *Enterococcus* genus, and on isolates of *E. coli, K. pneumoniae* and *P. aeruginosa.* (Table 1).

Since the MIC value of 128 µg/mL was assumed as the cut-off value above which to consider the compound inactive, CB1H was considered inactive against isolates of Gram-negative species (MICs > 128 µg/mL) and against some isolates of the *Enterococcus* genus (MICs = 256 µg/mL), regardless of their susceptibility to vancomycin, and weakly active against all other isolates considered in these experiments (MICs = 128 µg/mL).

According to the obtained results, although not exceptional, the MICs displayed by CB1H on MDR bacteria of Gram-positive species were lower than those of the reference antibiotics.

Consequently, since also water-soluble and clinically applicable in terms of administration feasibility and bioavailability, we thought that the pristine CB1H could nonetheless represent a promising template molecule to develop a new antimicrobial device, able to limit the spread of infections by MDR bacteria. Additionally, we thought that, if opportunely manipulated by means of nanotechnology, the intrinsic antibacterial effects of CB1H could be significantly improved.

In this regard, we opted to strategically nano-formulate it using a cationic copolymer (P7) previously developed by us, possessing a potent broad-spectrum bactericidal activity [17,18].

### 3.4. Entrapment of CB1H in Copolymer P7 (CB1H-P7 NPs)

The entrapment of small bioactive molecules in polymers, copolymers, and dendrimer nanoparticles provides several advantages, including improved solubility, stability, and efficacy, as well as reduced toxicity [13]. In particular, in cases of high drug-loading capacity (DL%), several equivalents of the entrapped drug can be released at the target site following the administration of a very low dosage of the complex polymer/drug. With this consideration, aiming at enhancing the antibacterial properties observed for CB1H and at obtaining a new antibacterial device possibly having a spectrum of action extended to Gram-negative pathogens, CB1H was entrapped in the cationic copolymer P7 we recently reported [17,18]. P7 was selected as the entrapping polymer agent mainly because, possessing potent antibacterial effects toward both Gram-positive and Gram-negative bacteria, it would allow us to obtain CB1H-loaded NPs more effective than pristine CB1H, and with an extended spectrum of action. Additionally, the cationic envelope provided by P7 to CB1H would promote the electrostatic interaction of CB1H-loaded NPs with the negatively charged bacterial surface, thus causing a significant accumulation of the pyrazole derivative on bacteria. Moreover, as for the most part of polymer-based cationic antibacterial agents, following the electrostatic interactions with the bacteria surface, P7 would modify the bacterial permeability by formation of pores, thus favoring the entrance of CB1H inside the bacterial cell [16,17,24,25].

For preparing our CB1H-loaded polymeric NPs, we considered the nano-coprecipitation process as the simplest and most reproducible one [19,26,27], and accordingly, CB1H-P7 NPs were prepared according to Figure 4.

Different from the usual procedure (consisting of using an organic phase to dissolve the polymer and the molecule to be trapped and an aqueous phase as non-solvent to precipitate the drug/polymer NPs), since P7 and CB1H were soluble in water, the aqueous phase was replaced by Et_2_O, in which both P7 and CB1H, as well as CB1H-P7 NPs, were insoluble. MeOH was chosen to solubilize both P7 and CB1H, affording a clear methanol solution. No surfactant or other additives, which in any future clinical use of CB1H-P7 NPs could be responsible for unwanted side reactions, were added, thus facilitating purification. The addition of the methanol solution of P7 and CB1H to the non-solvent under moderate stirring led to supersaturation and to the precipitation of CB1H-loaded P7 NPs, which were treated as described in detail in the experimental section. CB1H-P7 NPs were obtained as a white solid, which was stored on P_2_O_5_ in a dryer. From the organic solvents separated by the solid CB1H-P7 NPs, unentrapped CB1H was recovered, and its identity was confirmed by ATR-FTIR analysis (not reported). Figure 5a shows the ATR-FTIR spectrum of CB1H-P7 NPs, where many bands belonging to the entrapped CB1H can be detected, thus assessing the successful loading.

Figure 5b–d shows the magnification of a significant region of the ATR-FTIR spectra of CB1H-P7 NPs, CB1H and P7, showing the presence of many more analogies between the ATR-FTIR spectra of NPs and the pristine pyrazole, rather than between those of NPs and P7, thus forecasting a high drug-loading value.

### 3.5. Principal Components Analysis (PCA) of ATR-FTIR Data

Although just by observing the ATR-FTIR spectra in Figure 5 the presence of CB1H in the prepared CB1H-P7 NPs was unequivocable, we have further confirmation from processing the ATR-FTIR spectral data using the principal components analysis (PCA) [9,19,20,28,29], which allowed us to visualize the reciprocal positions which CB1H, P7 and CB1H-P7 NPs occupied in the scores plot of PC1 (explaining the 81.9% of variance) vs. PC2 (explaining the 17.2% of variance) (Figure 6).

All samples were well-separated both on PC1 and on PC2, and reciprocally located at score values which showed that the prepared CB1H-P7 NPs are structurally more like CB1H than P7, thus announcing a high content of the pyrazole in the prepared NPs. In particular, all compounds were located at positive score values on PC1, but CB1H-P7 NPs were located more distant from P7 than from CB1H, indicating that in its structure the chemical groups of CB1H are present to a high degree.

Additionally, while on PC2 both CB1H and CB1H-P7 NPs were located at negative scores, thus highlighting structural similarities, P7 was positioned at positive scores, thus validating the presence of CB1H in CB1H-P7 NPs. Again, the very high similarities in the spectra of CB1H and CB1H-loaded NPs, as well as the very different position assumed in the score plot by P7 in respect of those of CB1H and CB1H-P7 NPs closer each other, can be explained by assuming that CB1H was mainly adsorbed to the surface of P7, rather than encapsulated.

### 3.6. CB1H Contained in CB1H-P7 NPs, DL% and EE%

Table 2 collects all data related to the construction of CB1H calibration model, including the values of absorbance [A] determined for each CB1H concentration analyzed, the concentrations of CB1H (C_CB1H_) used for the UV-Vis analyses, the concentrations of CB1H predicted by the calibration model (C_CB1Hp_), the residuals, and the absolute error percentages.

The [A] _mean_ and C_CB1H_ (µg/mL) reported in Table 2 were used to develop the CB1H calibration model by least squares (LS) method, whose equation was Equation (1), reported both in the experimental section and visualized in Figure 7a, which shows the obtained linear regression curve.

The high value of the coefficient of determination (R^2^ = 0.9998) confirmed the linearity of calibration. Nevertheless, for further confirmation, we assessed the linearity and sensitivity of the developed calibration model, also verifying the statistical significance of its slope. To this end, we performed the analysis of variance (ANOVA) by the Fischer test, and the statistical significance was established at the *p*-value < 0.05. Equation (1) was utilized to obtain the concentrations of CB1H predicted by the model (C_CB1Hp_) for each sample analyzed. C_CB1Hp_ were graphed vs. C_B1H_, obtaining the regression curve expressing the correlation existing between the two sets of data (Figure 7b). R^2^ (0.9998), as well as the value of the correlation coefficient R (0.9999), confirmed a strong correlation between the real and the predicted concentrations of CB1H, thus establishing the goodness of fit of the model.

The developed calibration model was used to determine the C_CB1H_ in the prepared CB1H-P7 NPs, and the values of DL% and EE%. Particularly, four solutions of CB1H-P7 NPs at different concentrations were subjected to UV-Vis analysis performed in triplicate, obtaining four values of [A] _mean_ ± SD at 250 nm (Table 3), which were used to determine the related C_CB1H_ (µg/mL) by employing Equation (1). The obtained concentrations (Table 3) were used to calculate the related amounts of CB1H contained in the amount of CB1H-P7 NPs weighted for the analysis (3.442 mg) and those contained in the total amount of CB1H-P7 NPs obtained from the entrapment reaction (140.5 mg). The results were then expressed as means ± SD (column three in Table 3). The latter value allowed us to determine the values of DL%, EE%, the moles of CB1H loaded for mole of P7, and to estimate the Mn of CB1H-P7 NPs (Table 3).

The CB1H contained in the amount of CB1H-P7 NPs analyzed (3.442 mg) was 1.674 ± 0.0505 mg, thus establishing that the CB1H loaded in the total amount of CB1H-P7 NPs obtained was 68.33 ± 2.0614 mg. Consequently, the DL% was 48.6%, while EE% was 81.5. Accordingly, P7 was able to load 34.7 moles of CB1H per mole, thus granting the delivery of a great concentration of CB1H at the target site of action, administering a minimal dosage of the CB1H formulation. The DL% observed in this study was the highest we have observed, including previous studies encapsulating other bioactive pyrazole derivatives (BBB4 and CR232) in cationic dendrimers of fourth and fifth generation (G4K and G5K) respectively, implementing different methods [9,20]. However, while the EE% herein determined was much higher than that observed upon encapsulation of BBB4 [9], it was slightly lower than that observed more recently upon encapsulation of CR232 [20]. Nevertheless, considering the only existing study reporting on the encapsulation of a pyrazole derivative (AMDPC) by using a commercial copolymer (PEG-PLGA), in which the values of DL% and EE% are reported, data determined by the authors for their micelles (DL% = 1.28 and EE% = 64.3) were extraordinarily lower than those herein determined for our pyrazole formulation, obtained using a copolymer synthetized by us [30].

The value of DL% herein determined was useful to estimate the average molecular mass (Mn) of CB1H-P7 NPs.

Briefly, the value of DL% (48.6 ± 1.4%) established that in the total amount obtained of CB1H-P7 NPs (140.5 mg), there should be 68.33 ± 2.0614 mg (0.1837 ± 0.0037 mmol) of CB1H (MW = 371.9), and 72.17 mg (0.0053 mmol) of P7 (Mn = 13,719). Accordingly, the moles of CB1H loaded per mole of P7 were 34.7 ± 0.7. From these data, and applying Equation (4), the Mn of CB1H-P7 NPs was estimated and reported in Table 3.
Mn_CB1H-P7_ = Mn of P7 (13,719) + 34.7 ± 0.7 × MW of CB1H (371.9)(4)

### 3.7. Morphology and Dynamic Light Scattering (DLS) Analysis of Particles of P7 and CB1H-P7 NPs

The SEM images of P7 and CB1H-P7 NPs are shown in Figure 8a,b, respectively. Both the empty copolymer P7 (Figure 8a) and CB1H-P7 (Figure 8b) showed a spherical morphology, while the average particle size was 343 and 333 nm, respectively, as reported in Table 4.

Importantly, the SEM images established that, following the loading of CB1H on copolymer P7, the spherical shape was maintained. In this regard, a spherical morphology contributes to providing nanoparticles with a high surface area, which in the case of drug-loaded polymers typically determines a longer systemic retention time and a slow metabolism, which results in improved therapeutic effects [9,19,20,31,32,33].

Curiously, although the very high DL% following the loading of CB1H was different from what we observed in previous work [19,20], the size of the particles did not increase. On the contrary, a slight reduction of the average dimension of particles was observed, probably due to the sonication procedure performed on CB1H-P7 NPs during isolation and purification, which as reported can lead to variations in the particle size and in the particle size distribution [34]. Particularly, it was established that sonication can result in a reduction of the effective diameter of particles, in a variation of particle size distribution, and in a bimodal distribution of particles, with an increase in the proportion of the smallest particles, dependent on the sonication time [34,35].

Moreover, the SEM micrographs showed that while P7 NPs appeared to be tightly attached to each other to form a large aggregate, those of CB1H-P7 NPs appeared mainly well-separated from each other. In this case, the phenomenon can be due to the sonication procedure, which, in addition to having affected the size and the size distribution of particles of CB1H-P7, promoted their separation [35]. In this regard, it has been reported that sonication was found to be an effective method to break up inter-micellar associations, and to split large polymeric aggregates, present initially in the aqueous solutions of poly (ethylene oxide)-block-polyisoprene, poly (ethylene oxide)-block-polystyrene di-block copolymers, and a poly (ethylene oxide)-block-polyisoprene-block-poly (ethylene oxide) triblock copolymer [35].

The size (Z-ave, nm), polydispersity index (PDI) and Zeta potential (ζ-p) of both P7 and CB1H-P7 NPs were determined by DLS analyses, and the results have been reported in Table 4.

Figure 9a,b shows representative images of particle size distribution of the empty copolymer P7 (**a**) and of the CB1H-loaded copolymer (CB1H-P7 NPs) (**b**), while Figure 9c,d shows the ζ-p mean distribution of P7 (**c**) and of CB1-P7 NPs (**d**).

For P7, the average particle size was 303.4 nm and the mean PDI was 0.692, whereas for CB1H-P7 NPs, values of 142.9 nm (size) and 0.0.626 (PDI) were observed. Concerning PDI, the value observed for P7 and for CB1H-P7 NPs were very similar, establishing that neither the loading of CB1H nor the sonication procedure affected the PDI of materials. According to what was observed in the SEM micrograph, but in a significantly more marked way, the particle size of CB1H-loaded NPs was significantly lower than that of empty dendrimers, thus confirming the assumption made in the previous section and what was reported in previous studies [34,35]. Both for P7 and CB1H-P7 NPs, two dimensional families were observed, but as reported in the literature [34], sonication performed only on CB1H-P7 NPs has led to a remarkable shift in the bimodal size distribution toward lower values of dimensions, providing particles with sizes in the range of 100–200 nm. As widely reported, in addition to morphology, the size of nanoparticles strongly influences the distribution, cytotoxicity, targeting ability, and drug-release profile of a drug formulation [33], and biomedical applications require sizes smaller than 200 nm, with an optimum of 100–200 nm [33], similar to the dimensions observed for CB1H-P7 NPs prepared by us. Concerning ζ-p, as expected CB1H-P7 NPs maintained a positive ζ-*p* value similar to that of P7 (+36.7 vs. +35.5 mV), which, being higher than the critical value of ± 30 mV, assures good physical stability of the formulation in a water solution [36]. Curiously, while the ζ-p distribution of P7 was mono-modal, that of CB1H-P7 NPs was bimodal, due to the presence of two types of positively charged ingredients, the copolymeric reservoir and the loaded hydrochloride CB1H, probably mainly adsorbed rather than encapsulated, as previously discussed in Section 3.6.

### 3.8. Release Profile of CB1H from CB1H-P7 NPs Determined by In Vitro Studies

The release profile of CB1H from CB1H-P7 NPs was studied in PBS receptor medium (pH = 7.4), performing the dialysis method as recently described [9,19,20]. The amount of CB1H which was released was quantified by analyzing aliquots withdrawn at fixed points for 24 h, by UV-Vis analyses, and using Equation (1).

Determinations were made in triplicate, and results were expressed as CB1H cumulative release percentages (CR %), which are given for each time by Equation (5),
(5)$$CR\;\left(\%\right)=\frac{CB1H\left(t\right)}{CB1H\;\left(NPs\right)}\times 100\;$$
where CB1H (t) is the amount of CB1H released at (t) incubation time, while CB1H (NPs) is the total CB1H entrapped in the weight of CB1H-P7 NPs analyzed, according to the computed DL%.

Figure 10 shows the CB1H release profile obtained by reporting the determined CR% values vs. the incubation times in a dispersion graph.

As in our previous works where we determined the release profiles of drug-loaded nanomaterials prepared by us, designed to develop new antibacterial agents [9,19,20], the release profile of CB1H was monitored for 24 h to have data compatible with the values of MICs, whose determination will be the scope of our subsequent article. According to EUCAST protocols [17], antibacterial activities must be established by determining the MIC values after 18–24 h. Consequently, to know the concentration of CB1H that has contributed to the MICs that we observed, we should know both the amount of CB1H loaded into CB1H-P7 NPs (DL%) and the amount of CB1H released at 24 h. Figure 10 showed that practically all CB1H was finally released and was released within the first 12 h.

Particularly, the release of CB1H from CB1H-P7 NPs was characterized by a slow release in the first 4 h, followed by an eight-hour phase of a faster and almost linear release, leading to a release of 60.2%. In the following 12 h, we can observe a plateau during which only an additional 5.6% of CB1H was released.

The kinetics and the main mechanisms which govern the release of CB1H from CB1H-P7 NPs were determined by fitting data of the CB1H CR% curve with the equations of zero-order, first-order, Hixson–Crowell, Higuchi, Korsmeyer–Peppas, and Weibull mathematical models [20,37,38,39,40]. The highest value of the coefficient of determination (R^2^) of the equations of the tendency lines, related to the dispersion graphs obtained, was considered as the parameter to determine which model better fits the release data. R^2^ values were 0.7743 (zero-order), 0.8358 (first-order), 0.9287 (Korsmeyer–Peppas model), 0.8140 (Hixson–Crowell model), 0.8845 (Higuchi model), and 0.9397 (Weibull model), thus establishing that the release of CB1H from the developed delivery system best fit with the Weibull mathematical model (Figure 11).

The equation expressing the Weibull kinetic model is the following Equation (6):(6)$$LnLn\frac{C_{0}}{C_{0}-C_{t}}=\beta Lnt+Ln\alpha \;\;\;$$
where *C_t_* is the concentration of drug release at time *t*, *C*_0_ is the initial concentration of drug present in the nanocomposite system, *t* is the time, *β* is the shape parameter of the dissolution curve, and α is the scale parameter.

According to the linear regression shown in Figure 11 and Equation (6), the slope of the regression corresponds to the value of *β*, while the value of *α* can be calculated by the value of intercept (*Lnα*). Values of *β* < 0.75 imply that the drug release is governed by diffusion mechanisms [20], while values in the range 0.75–1.0 suggest a combined mechanism frequently encountered in the release studies [20]. *β* = 1 indicates a first-order release, in which the drug concentration gradient in the dissolution medium controls the rate of its release. Finally, *β* > 1, as in this case, indicates that a complex mechanism governs the release process [20].

### 3.9. Potentiometric Titrations

P7 has primary amine groups, while CB1H-P7 NPs have primary amine groups provided by P7 and secondary amine groups provided by CB1H. The protonation of both types of basic groups is reversible, depending on the pH value of the environment. The protonation of the amine groups of developed NPs is essential for having a positively charged surface, which is crucial for allowing effective electrostatic interactions with the bacterial surface, thus promoting significant antibacterial activity. Therefore, we evaluated whether CB1H-P7 NPs were protonated in the physiological pH range of 4.5–7.5. To obtain this information, the potentiometric titration of CB1H-P7 NPs, and for comparison purposes that of P7, has been carried out according to Benns et al. [16,21].

By reporting the assessed pH values vs. the volumes of HCl 0.1 N added (0.2 mL), the titration curves of CB1H-P7 NPs and P7 were obtained (Figure 12a, red and purple lines, respectively).

Subsequently, from titration data, the dpH/dV values were computed and graphed vs. those of the corresponding volumes of HCl 0.1 N, obtaining the first derivative lines of the titration curves (Figure 12b). The maxima of the first derivative curves corresponded to the titration’s endpoints. Interestingly, for both samples more than one maximum was observed, due to the presence of different types of amine groups. While for P7, which contains only primary amine groups, two maxima were observed (probably since it is a random copolymer containing basic blocks of different dimension), for CB1H-P7 NPs three maxima were observed, due to the presence of the secondary amine groups provided by CB1H. According to the obtained results, upon three protonation steps at pH = 8.00, 7.00, and 4.90, in the physiological pH range (4.8–8, depending on whether it is measured), CB1H-P7 NPs should be completely protonated, assuring in vivo applications, and a cationic chemical structure capable of strong electrostatic interactions with negatively charged bacterial surface.

## 4. Conclusions and Perspectives

In this study, starting from a basic water-insoluble pyrazole derivative (CB1) endowed with various pharmacological activities, a water-soluble hydrochloride (CB1H), potentially in vivo-administrable, was synthetized, characterized, and preliminarily evaluated as a possible antibacterial agent. Specifically, in the form of a hydrophilic small molecule, CB1H did not display activity on isolates of Gram-negative species (MICs > 128 µg/mL), while it exhibited weak antibacterial effects on MDR isolates of Gram-positive species, which, even if not exceptional, were higher than those displayed by the commonly used available antibiotics. So, aiming at improving such activity and at extending the antibacterial spectrum of CB1H to Gram-negative pathogens via a nanotechnological approach, in this first work based on CB1, it has been strategically formulated in nanoparticles using a cationic copolymer (P7) endowed with a very potent broad-spectrum bactericidal activity. Once we obtained the CB1-loaded P7-based NPs, we performed a rigorous characterization to determine the chemical composition of the obtained CB1H-P7 NPs by acquiring the ATR-FTIR spectra of CB1H, P7, and CB1H-P7 NPs, and processing the matrix of the spectral data of all samples by PCA, which predicted a high value of DL%. In addition to the EE%, the DL% was obtained by UV-Vis analyses, whose values confirmed the prediction of PCA and were used to estimate the Mn of CB1H-P7 NPs. Particularly, the DL% (48.6%) was the highest of those obtained by us encapsulating other pyrazole derivatives using dendrimer NPs. We think that due to the water-solubility of CB1H, different from the previous cases where NPs acted both as solubilizing and entrapping agent, P7 was required to act only as an entrapping agent, thus improving its loading capacity. The DL% determined this time was 27-fold higher than that recently reported of micellar NPs obtained by entrapping a pyrazole derivative in commercial PEG-PLGA. The in vitro release study showed a triphasic release profile governed by Weibull kinetics, and a release of 66% of CB1H after 24 h. The DLS experiments established that the CB1H-P7 particles were nano-sized, with a bimodal distribution and dimensions in the range 100–200 nm, assuring low systemic toxicity. Micrographs obtained by SEM showed a spherical morphology, which means a high surface area, and which typically translates to high systemic residence time and bio-efficiency, and no presence of aggregates. Additionally, the ζ-*p* value was positive and higher than 30 mV, thus ensuring high stability in aqueous solution and no tendency to re-form aggregates. Titration experiments demonstrated the presence of nitrogen atoms, which proved to remain protonated in the physiological pH range.

By means of a nanotechnological manipulation using P7, CB1H has been converted to potentially clinically usable P7-based NPs, which, based on the established potent broad-spectrum antibacterial activity of P7, are promising for developing a new potent antibacterial agent whose biological investigations will be the subject of our next work. The novelty and originality of the research reported herein consists of preparing cationic water-soluble NPs loaded with a hydrophilic pyrazole derivative, achieving a very high DL% never reported previously for pyrazole-loaded polymer systems, and a particle size suitable for biomedical applications, performing a simple, fast, one-step coprecipitation process, not requiring additional mechanical devices. Particularly, we have successfully formulated hydrophilic CB1H using the hydrophilic P7 without resorting to the most widely used method, known as the double emulsion solvent evaporation method (DE-SE) requiring an initial shear mixing and then a complex multi-step manufacturing process, where many variables can affect the quality of fabricated particles, such as EE%, DL%, particle size, and release profile, which in our case were excellent.

## Data Availability

All data concerning this study are contained in the present manuscript or in previous articles whose references have been provided.

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
