# Peer review of "Synthesis and Characterization of Pyrazole-Enriched Cationic Nanoparticles as New Promising Antibacterial Agent by Mutual Cooperation"

_nanomaterials, 2022, doi:10.3390/nano12071215_

Round 1
Reviewer 1 Report
Comments on the MS by alfei et al.
The MS untitled Synthesis and Characterization of Pyrazole-Enriched Cationic Nanoparticles as New Promising Antibacterial Agent focus a relevant topic because the scarcity in antimicrobial agents mainly due to bacterial strains resistance to the most antimicrobial agents.
Introduction:
“Among the different nanocarriers, nanoparticles including both nanocapsules 108 and nanospheres have emerged as a very promising strategy.” Provide a reference.
Introduction needs attention because in parts results are being provided Example: Once CB1H was obtained, preliminary antimicrobial investigations herein re-100 ported were performed with CB1H on representative MDR isolates of Gram-nega-101……
It would be interesting to provide the hypothesis or hypotheses under testing.
Consider reduce the number of sections/sub-sections in Material and methods. Some are just a sentence Table 1. “Experimental data for the “ No needed
In data of table 2 Provide standard deviation
The results and discussion section needs attention. Too many sections that make difficult to follow main findings.
Author Response
Comments on the MS by alfei et al.
The MS untitled Synthesis and Characterization of Pyrazole-Enriched Cationic Nanoparticles as New Promising Antibacterial Agent focus a relevant topic because the scarcity in antimicrobial agents mainly due to bacterial strains resistance to the most antimicrobial agents.
Introduction:
“Among the different nanocarriers, nanoparticles including both nanocapsules 108 and nanospheres have emerged as a very promising strategy.” Provide a reference.
As requested by the Reviewer a reference has been provided. Please see line 108 and lines 888-890.
Introduction needs attention because in parts results are being provided Example: Once CB1H was obtained, preliminary antimicrobial investigations herein re-100 ported were performed with CB1H on representative MDR isolates of Gram-nega-101……
It would be interesting to provide the hypothesis or hypotheses under testing.
We thank the Reviewer for his suggestion. The sentence indicated by the Reviewer has been reformulated and the hypothesis under testing has been included (lines 97-101).
Consider reduce the number of sections/sub-sections in Material and methods. Some are just a sentence Table 1. “Experimental data for the “ No needed
Following the suggestions of the Reviewer, in Material and Methods, all the sub-sections have been removed except for Section 2.3, and Sections have been reduced. Concerning Table 1, it makes part of Results and Discussion Section and not of Material and Methods one. Anyway, as suggested by the Reviewer Table 1 has been removed.
In data of table 2 Provide standard deviation
We would like to kindly point out to the Reviewer that in microbiology the MICs data are never reported with the standard deviation (SD) because the reported value is not an average of a series of measured values, but it is a modal value or the value that has been observed most frequently. However, in order not to arouse such doubts/comments, we had already explained in the original version of the manuscript the reason for the absence of SD values. In fact, if the Reviewer takes into consideration the footnotes of Table 2 (now Table 1), he will find the following note referring to the MIC values and the absence of SD values:
$ experiments were performed in triplicate, the concordance degree was 3/3, and ± SD was zero.
The results and discussion section needs attention. Too many sections that make difficult to follow main findings. The proposed manuscript is interesting, and I suggest its publication after some corrections.
We thank a lot the Reviewer for his appreciation. According to his suggestion the sections and sub-sections of the Results and Discussion section have been reduced.
Reviewer 2 Report
The submitted manuscript describes and discusses the results of an original research project carried out to synthesis and characterization of pyrazole-enriched cationic nanoparticles as new promising antibacterial agent by mutual cooperation. The manuscript describes and discusses logically designed experiments and presents results that are expected to be of large interest for the scientific community. It is an interesting study with an interesting approach. The paper in the whole is well designed and results sound. Nevertheless, the manuscript needs a major revision:
- In the introduction part should be more highlighted the main aim of the paper, and additionally, what is the novelty of carried research work.
- How do the Authors select the analytes? The rational of the choice of the selected biologically active compounds studied is missing and should be clearly discussed.
- Quality of the figures must be improved.
Author Response
The submitted manuscript describes and discusses the results of an original research project carried out to synthesis and characterization of pyrazole-enriched cationic nanoparticles as new promising antibacterial agent by mutual cooperation. The manuscript describes and discusses logically designed experiments and presents results that are expected to be of large interest for the scientific community. It is an interesting study with an interesting approach. The paper in the whole is well designed and results sound. Nevertheless, the manuscript needs a major revision:
In the introduction part should be more highlighted the main aim of the paper, and additionally, what is the novelty of carried research work.
We make kindly note that the present paper is the first part of a wider research work whose final goal has been specified in lines 70-79. Anyway, as suggested by the Reviewer, although the main aim of the present work was already explained in lines 109-110, it has been further highlighted in lines 97-101.
Concerning the novelty of the carried research work, we agree with the Reviewer that it must be highlighted. Anyway, we have preferred to evidence the novelty of our work in the Conclusions (lines 826-837). We hope that the Reviewer would accept such deviation from his suggestion.
How do the Authors select the analytes? The rational of the choice of the selected biologically active compounds studied is missing and should be clearly discussed.
We make kindly note to the Reviewer that the rational for the choice of the pyrazole derivative was already widely explained and discussed in lines 75-96, while the rational for the choice of P7, already explained in lines 111-112, has been better explained in the new lines 112-121.
Quality of the figures must be improved.
All Figures have been provided with resolution higher than that required by Nanomaterials. Anyway, their quality has been further improved by working on their contrast and luminosity.
Round 2
Reviewer 1 Report
The authors adressed the comments made on the MS. So, it can be aceepted for publication